# Super-Resolution Analysis of the Origins of the Elementary Events of ER Calcium Release in Dorsal Root Ganglion Neurons

**DOI:** 10.3390/cells13010038

**Published:** 2023-12-23

**Authors:** Miriam E. Hurley, Shihab S. Shah, Thomas M. D. Sheard, Hannah M. Kirton, Derek S. Steele, Nikita Gamper, Izzy Jayasinghe

**Affiliations:** 1School of Biomedical Sciences, Faculty of Biological Sciences, University of Leeds, Leeds LS2 9JT, UK; 2School of Biosciences, Faculty of Science, The University of Sheffield, Sheffield S10 2TN, UK; 3EMBL Australia Node in Single Molecule Science, School of Biomedical Science, University of New South Wales, Kensington, Sydney 2052, Australia

**Keywords:** dorsal root ganglion neurons, correlative microscopy, dSTORM, expansion microscopy, ryanodine receptor, inositol 1,4,5-trisphosphate receptor, calcium signalling

## Abstract

Coordinated events of calcium (Ca^2+^) released from the endoplasmic reticulum (ER) are key second messengers in excitable cells. In pain-sensing dorsal root ganglion (DRG) neurons, these events can be observed as Ca^2+^ sparks, produced by a combination of ryanodine receptors (RyR) and inositol 1,4,5-triphosphate receptors (IP3R1). These microscopic signals offer the neuronal cells with a possible means of modulating the subplasmalemmal Ca^2+^ handling, initiating vesicular exocytosis. With super-resolution dSTORM and expansion microscopies, we visualised the nanoscale distributions of both RyR and IP3R1 that featured loosely organised clusters in the subplasmalemmal regions of cultured rat DRG somata. We adapted a novel correlative microscopy protocol to examine the nanoscale patterns of RyR and IP3R1 in the locality of each Ca^2+^ spark. We found that most subplasmalemmal sparks correlated with relatively small groups of RyR whilst larger sparks were often associated with larger groups of IP3R1. These data also showed spontaneous Ca^2+^ sparks in <30% of the subplasmalemmal cell area but consisted of both these channel species at a 3.8–5 times higher density than in nonactive regions of the cell. Taken together, these observations reveal distinct patterns and length scales of RyR and IP3R1 co-clustering at contact sites between the ER and the surface plasmalemma that encode the positions and the quantity of Ca^2+^ released at each Ca^2+^ spark.

## 1. Introduction

Ca^2+^ released from internal compartments such as the endoplasmic reticulum (ER), lysosomes, Golgi and mitochondria is a ubiquitous type of fast, intercellular second messenger signal in eukaryotic cells. Most notably, rapid cytoplasmic Ca^2+^ transients form the fundamental excitation–contraction mechanism in muscle [1]. Ca^2+^ can also act as a trigger for a host of other cellular responses such as gene transcription [2], exocytosis [3,4], metabolic switching [5], cell motility [6], autophagy [7] and programmed cell death (see review [8]). In excitable cell types, multiple Ca^2+^ handling proteins are commonly organised within nanoscale signalling domains, often consisting of ryanodine receptors (RyRs) or inositol 1,4,5-triphosphate receptors (IP3Rs) that can engage in coordinated and/or propagating Ca^2+^ release from the ER. The most observed elementary Ca^2+^ release events within cells are ‘Ca^2+^ sparks’ in muscle, caused by the activation of RyR [9,10]. IP3Rs can also produce elementary Ca^2+^ release events, observed as ‘Ca^2+^ puffs’ in the presence of IP3 [11], although in most cell systems they produce detectable Ca^2+^ release in coordination with RyRs.

Mammalian neurons can express any or all RyRs, types 1 to 3 [12], along with IP3R types 1 and 3 [13]. Ca^2+^ released from these mechanisms in neurons, particularly along neurites, is implicated in synaptic plasticity and the modulation of excitability in the postsynaptic membranes [14]. In pathologies and injuries, RyR-mediated Ca^2+^ release is also implicated in the mechanisms leading to excitotoxicity of neurons [15,16].

In one of the few documented studies of Ca^2+^ sparks in dorsal root ganglion (DRG) neurons to date, Ouyang et al. show that pharmacologically induced Ca^2+^ sparks alone can mobilise and induce exocytosis of pain-related bioactive cargo such as calcitonin gene-related peptide and substance P [4]. IP3R1 has been found as a trigger for the Ca^2+^-activated Cl^-^ channel, anoctamin 1 (ANO1), in DRG [17]. In turn, IP3R1 forms a critical link in the functional coupling between ANO1 and nociceptive transient receptor potential vanilloid 1 (TRPV1) channels, all colocalised in plasmalemma-ER junctional complexes [18]. Finally, Ca^2+^ sparks are also implicated in the mobility, and in turn, the stimulation of luminal ER Ca^2+^ sensors, stromal interaction molecules (STIMs) [19]. Activation of store-operated Ca^2+^ entry via the interaction of STIM1 and plasmalemmal Orai1 channels can have a knock-on effect on the excitability of DRG neurons by providing Ca^2+^ for excitatory G-protein-coupled receptor (GPCR) signalling [20], and could also contribute to noxious cold sensation [21].

The role of RyR in pain perception specifically is less clear. There are conflicting reports that RyR expression in DRG neurons either is limited to types 1 and 3 only [14], is limited to type 3 only (in DRG microsomes) [22] or consists of all three isoforms [16]. A higher density of RyR, particularly RyR3 in the subplasmalemmal ER network, nevertheless [4] could sustain this store-driven modulation of nociceptive signals.

Visualisation of different RyRs in various cell types has been considerably improved by us and others over the past decade with the use of various super-resolution microscopy methods and revealed the organisation of RyRs to follow distinct clusters or arrays (see review [23]). Whilst this technology has not been extended to RyRs in DRG, confocal images by Ouyang et al. suggest intricate, subplasmalemmal organisation of RyR clusters present in the DRG somata [4]. Nanoscale clusters of IP3R1 have been visualised with STORM [18]. The lack of either correlative imaging protocols or genetically tagged animal models of fluorescent (e.g., green fluorescent protein-fused) RyR or IP3R has meant that the spatial relationship between subplasmalemmal Ca^2+^ sparks and the underlying channel organisation could not be probed directly until now.

Here, we present the leveraging of super-resolution expansion microscopy (ExM) and direct stochastic optical reconstruction microscopy (dSTORM) to visualise the nanoscale clustering patterns of the dominant Ca^2+^ release channels, RyR3 and IP3R1, in primary DRG cultures. To enable the novel application in DRG neurons, we have adapted a recently developed correlative imaging protocol to spatially correlate Ca^2+^ sparks (and spark-like events) with the subplasmalemmal patterns of RyR3 and IP3R1 organisation resolved with dSTORM.

## 2. Materials and Methods

### 2.1. Microscope Setup

All experiments were performed with a Nikon TE2000 (Nikon; Tokyo, Japan) modified to enable total internal reflection fluorescence (TIRF) dual-colour imaging with 488 nm and 671 nm excitations on a ~15 μm × 15 μm illumination field (as detailed previously [24]). The full list of specifications and settings can be found in the Appendix A. Emitted light was recorded onto a Zyla 5.5 scientific complementary metal–oxide–semiconductor camera (sCMOS; Andor, Belfast, UK). Raw image series for Ca^2+^ imaging were acquired using open-source Python Microscopy Environment [25] (PyME) software.

### 2.2. Isolation and Culture of DRG

Experiments were performed according to the UK Animals (Scientific Procedures) Act of 1986 and with UK Home Office approval and local ethical approval. Neonatal male Wistar rats (aged 6–10 days) were euthanised according to Schedule 1 outlined by the UK Home Office, via an overdose of isoflurane, an inhalational anaesthetic, followed by cervical dislocation. DRGs were dissected from the spine and incubated for 15 min (37 °C humidified chamber, air supplemented with 5% CO_2_) in Mg^2+^- and Ca^2+^-free Hanks’ balanced salt solution containing 10 mg/mL dispase (ThermoFisher Scientific, Waltham, MA, USA) and 1 mg/mL Type 1A collagenase (Sigma-Aldrich, Saint Louis, MO, USA).

Enzymatic reaction was halted by the addition of ice-cold DMEM pre-supplemented with GlutaMAX, 10% foetal bovine serum, 5% penicillin and 50 mg/mL of streptomycin (Sigma-Aldrich) to inactivate the enzymes before cells were centrifuged (800 rpm; 5 min; 4 °C) and resuspended in 800 μL of the fresh, pre-supplemented DMEM solution. These cell suspensions contained DRG neuronal cell bodies as well as glial cells.

Cells were plated onto a 50 mg/mL laminin and 0.001% poly-D-lysine-coated 500 mm square gridded imaging dish, with #1.5H glass coverslip (Ibidi, Fitchburg, WI, USA), and incubated within a humidified chamber at 37 °C with 5% CO_2_ concentration for 4 h. Dishes were then flooded with additional pre-supplemented DMEM and maintained for a further 48 h under the same conditions as described previously [26]. We did not add any further inhibitors of glial cells; therefore, these DRG neurons were sustained cocultures throughout the experiments presented in this paper.

### 2.3. Ca^2+^ Spark Imaging

After 48 h of culturing, DMEM was replaced with Tyrode’s solution, containing (in mM) 160 NaCl, 1 MgCl_2_, 2.5 KCl, 10 HEPES and 10 glucose, at pH 7.4 with added 0.75 mM CaCl_2_.

DRG cocultures were loaded with Fluo-4 AM fluorescent Ca^2+^ indicator (ThermoFisher Scientific) and immersed in a Tyrode’s solution containing 5 mM CaCl_2_ at pH 7.4. The dishes were clamped securely onto the stage of the TIRF microscope system such that the grid was aligned with the straight edges of the camera’s field of view, under brightfield illumination. DRG cells forming a substantial footprint were illuminated with a 488 nm laser. The local changes in the Fluo-4 fluorescence were recorded at a 100 ms/frame rate with the outline of the cell’s contact patch with the grid included within the imaging frame. See detailed protocol in the Appendix A.

During Ca^2+^ spark imaging, cells were immersed in Tyrode’s solution either alone, or with 5 mM caffeine or 250 nM bradykinin added to activate the RyR and IP3R1 receptors, respectively.

### 2.4. dSTORM Imaging and Primary Processing

Following the imaging of Ca^2+^ sparks, cells were fixed in 2% paraformaldehyde (Sigma-Aldrich; *w*/*v* in phosphate-buffered saline; PBS) for 10 min at room temperature (RT), prior to 4 washes of 10 min in PBS. Cells then underwent immunofluorescence labelling; this consisted of a 10 min incubation with 0.1% Triton X-100 followed by a 60 min incubation with PBS containing 10% NGS (normal goat serum), both at room temperature (RT). Cells were incubated at 4 °C overnight with either rabbit anti-RyR3 (AB9082; Sigma-Aldrich) or rabbit anti-IP3R1 (PA1-901; ThermoFisher Scientific) primary antibody diluted 1:200 in incubation solution (*w*/*v* or *v*/*v*; 0.05% NaN_3_, 2% BSA, 2% NGS and 0.05% Triton X-100 dissolved in PBS), and washed 3 times in PBS at RT for 20 min, followed by a 2 h incubation at RT with antirabbit Alexa 660 (A21073; Invitrogen, Waltham, MA, USA) diluted 1:200 in incubation solution. Both the anti-RyR3 [27] and anti-IP3R1 [17,18] primary antibodies have previously been extensively characterised for use in DRG.

For dSTORM acquisition, dishes were returned to the stage of the TIRF microscope and securely clamped, and the grid coordinates recorded from Ca^2+^ imaging, alongside the brightfield illumination, were utilised to relocate the corresponding cells. Cells were exposed to the glycerol-based mounting medium consisting of 10–20 mM cysteamine, 1.5 mM Na_2_SO_4_ and 10× PBS, pH 7.0. The 671 nm laser was focused at a supra-critical angle onto the field of view and a time series of dSTORM events was acquired at 50 ms/frame for 20,000–30,000 frames. The event positions were then rendered into a 16-bit greyscale TIFF image with a pixel scaling of 5 nm/pixel using an algorithm based on Delaunay triangularisation.

### 2.5. Correlative and Quantitative Image Analysis

The full image analysis protocol consisting of the registration of the Ca^2+^ spark data to the super-resolution data and the localisation of Ca^2+^ sparks has been published previously [24].

#### 2.5.1. Ca^2+^ Spark Detection

The Ca^2+^ spark localisation tool ImageJ plugin software ‘xyspark’ [28] was used for background estimation, detection and localisation of individual Ca^2+^ sparks. Output of this analysis is a list of x, y and t coordinates of each spark along with their full width at half maximum (FWHM) estimated using the Gaussian fit, coefficient of determination *R*^2^ and amplitude estimated as *F*/*F*_0_, where *F*_0_ was the estimate of the baseline level of the Ca^2+^ indicator fluorescence in a local cytoplasmic region and *F* was the fluorescence intensity value at the peak of the spark. Only sparks with 0.5 μm ≤ FWHM ≤ 2.0 μm and an *R*^2^ value ≥ 0.5 were filtered and retained for further analysis.

#### 2.5.2. RyR3 and IP3R1 Puncta Localisation

The punctate RyR3 and IP3R labelling densities in the rendered dSTORM images were detected using a centroid detection algorithm available through PyME and described previously [29]. The list of coordinates output from this analysis was used for the correlative analysis of the RyR3 or IP3R channels and the analysis of neighbour distances between local channels underneath the spark footprints.

#### 2.5.3. Image Correlation and Alignment of Discretised Ca^2+^ Sparks and RyR3 Puncta

We used an image alignment pipeline code to upscale the Ca^2+^ images and align the coordinates of the Ca^2+^ sparks to the super-resolution image, as described recently [24]. Briefly, ~10 consecutive frames from the Ca^2+^ time series were averaged to produce a low-noise, diffraction-limited image of the cell. This image was taken as a reference image of the cell’s relative position against the cell boundaries. The code upscaled the low-noise Ca^2+^ image to match the pixel scaling of the dSTORM image. It then required the user to manually align the Ca^2+^ image against the dSTORM image, using the cell outline as a guide. The x and y shift coordinates used for this alignment were then used as a starting point to perform an automated fine alignment through cross-correlation of the images. The shift coordinates determined through the fine alignments were then applied to the Ca^2+^ spark coordinates to align them against the maps of RyR3 or IP3R1 puncta. See [24] for details.

#### 2.5.4. Local Sampling of RyR3 or IP3R1 Organisation Using Ca^2+^ Spark Footprint

RyR3 or IP3R1 centroids which were located in proximity to a circular window with a diameter equal to the FWHM of the spark were included in the correlative analysis for each spark. In correlation to the puncta count beneath each spark against ‘spark mass’, the latter was calculated as the product of the spark amplitude (*F*/*F*_0_), FWHM [29] and conversion factor 1.206, and was one of the default output parameters of the xyspark software [30]. The nearest neighbour distance value for each spark sampled represented the average of the distance from each punctum centroid to that of its nearest neighbour.

### 2.6. Enhanced Expansion Microscopy

DRG cocultures that did not undergo the above correlative imaging protocol were fixed in 2% paraformaldehyde (Sigma-Aldrich; *w*/*v* in phosphate-buffered saline; PBS) for 10 min at RT, prior to 4 washes of 10 min in PBS. Cells underwent immunofluorescence labelling following the above methodology, and were incubated at 4 °C overnight with either rabbit anti-RyR3 (AB9082; Sigma-Aldrich) or rabbit anti-IP3R1 (PA1-901; ThermoFisher Scientific) primary antibody diluted 1:200 in incubation solution, followed by a 2 h incubation at RT with antirabbit Alexa 488 (A11008; Invitrogen) diluted 1:200 in incubation solution. Samples then underwent the 10× enhanced expansion microscopy (EExM) protocol detailed by Sheard et al. [31]. DRGs, distinguishable from satellite glial cells by their distinctive shapes and size, were imaged on an inverted LSM880 with Airyscan (Carl Zeiss, Jena, Germany) using a Plan-Apochromat 40× 1.3 NA objective with 0.21 mm working distance and a Plan-Apochromat 20× 0.8 NA objective lens. Image processing, to convolve the EExM image with a confocal point spread function (PSF), was undertaken in Image J version 2.14 (Fiji).

### 2.7. Statistics

Comparison of RyR3 or IP3R count with the Ca^2+^ spark properties of the DRG was undertaken using a repeated measures ANOVA with Tukey post hoc analysis (*p* < 0.05). All statistical tests were performed in GraphPad Prism.

## 3. Results

### 3.1. Optical Visualisation of Fast Ca^2+^ Sparks and Their Molecular Origins in DRG

We used total internal reflection fluorescence (TIRF) microscopy of DRG and satellite glial cell primary cocultures as our primary assay for studying the subplasmalemmal Ca^2+^ sparks in sensory neurons. When loaded with cytoplasmic Ca^2+^ indicator Fluo-4 AM, the somata of the DRG appeared in diffraction-limited TIRF images as large and often circular regions with heterogeneous fluorescence (Figure 1A). Under fast camera acquisitions under 5 mM extracellular [Ca^2+^]_o_, transient and localised Ca^2+^ sparks were observed (Figure 1B). One of the key observations made when cells were independently exposed to the proinflammatory G-protein-coupled receptor ligand, bradykinin or caffeine (stimulants of IP3R1 and RyR3, respectively) was the distinct increase in the mean of the integrated intensity of the sparks (termed ‘spark mass’) by ~70% and ~63%, respectively (Figure 1C-main). There was no measurable change to the width of the sparks (Figure 1C-inset), leading us to the hypothesis that sparks were spatially encoded by relatively fixed numbers of Ca^2+^ release channels. Figure 1D is a schematic drawing of the topological complexities that we hypothesise in the ER network of DRG somata. Observations by Ouyang et al. [4] (red symbols in the magnified view of the schematic in Figure 1E) included a higher density of the channel localised at the subplasmalemmal regions compared with the deeper interior. This prompted us to focus our investigation near the surface of the DRG somata.

The nanoscale distribution of IP3R1 was visualised using a 10× EExM protocol on IP3R1 immunofluorescence labelling in isolated and cultured DRG somata. Figure 1F shows an overview of a 10× ExM image acquired near the basal surface of a soma with a 0.8 NA objective lens (at an effective resolution of ~39 nm). The higher-resolution view (image acquired in Airyscan mode with a 1.3 NA objective; a protocol termed 10× EExM which offers an in-plane resolution of ~15 nm [31]) revealed a spatially heterogeneous, punctate morphology. This morphology was qualitatively similar to the pattern observed in dSTORM image data (Figure 1H) at a resolution of ~30 nm (see Appendix A for the resolution test). Histograms of the puncta size showed very similar distributions between 10× EExM and dSTORM (see Appendix A). RyR3 immunofluorescence labelling also followed punctate morphology in the two comparison points of EExM (Figure 1I,J). However, sub-micron-scale domains featured noticeable clustering of RyR3 (dashed circles) in both ExM and dSTORM images of equivalent samples (Figure 1J,K).

### 3.2. An Experimental Pipeline to Correlate Ca^2+^ Sparks with Nanoscale Organisation of RyR3 and IP3R1

To further examine the spatial relationship between these punctate densities of subplasmalemmal RyR3 and IP3R1, we adapted a recently developed experimental pipeline that spatially correlates fast Ca^2+^ sparks with the super-resolution maps of the underlying proteins [24,29] (see schematic in Figure 2A). The post hoc analysis included the rescaling and a semiautomated alignment of a time-averaged facsimile of the Ca^2+^ time series (green) against the dSTORM image (magenta), primarily using the unique cell outline of the soma to overlay the Ca^2+^ sparks with the underlying Ca^2+^ channel map (Figure 2B,C). A unique view enabled from this approach is the 2D overlay of the spark positions (shown over a 10 s time window) and the super-resolution dSTORM maps of RyR3 (Figure 2D) and IP3R1 (Figure 2E). Overlays such as these reveal not only coinciding clusters and sparks, but also spatial heterogeneity in the spontaneous spark activity.

### 3.3. Local Sampling of RyR3 and IP3R1 Channels Underlying Ca^2+^ Sparks

The correlative datasets, following rescaling and alignment, allowed us to sample the RyR3 or IP3R1 channel organisation patterns in the locality of the Ca^2+^ sparks recorded. Figure 3A shows a frame from a Ca^2+^ spark TIRF image series and the aligned dSTORM image of RyR3. The magnified views of the region of interest show the RyR3 puncta that occupy the locality of the Ca^2+^ spark. For further statistical analysis, the centroids and the shape descriptors of the Ca^2+^ sparks, along with the centroids of the RyR3 or IP3R1 puncta, were localised into two series of coordinates and registered against each other. This allowed the puncta and clusters within the ‘spark footprint’, determined by a circular window (green dashed lines in Figure 3B) centring the spark’s centroid and diameter equal to the spark’s full width at half maximum (FWHM), to be counted and sampled on a spark-by-spark basis.

This local sampling of the nanoscale structure permitted a direct correlation of the spark mass (the integrated signal in a Ca^2+^ spark) with the total number of puncta locally. In scattergrams, we observed a steep gradient between the spark mass and the RyR3 puncta (Figure 3C) in a range of spark masses varying across at least three orders of magnitude. Over half of the sparks sampled in this analysis consisted of relatively small numbers of local RyR3 (≤10 puncta; mean ± SD of 19.4 ± 26.6). Across a similar range of spark mass values considered, we observed an approximately two-fold higher mean and spread in the total number of IP3R1 puncta sampled within the spark footprint (mean ± SD of 34.8 ± 41.79; Figure 3D, Appendix A). The correlation was comparatively weaker between spark mass and the IP3R1 puncta counts in sparks with smaller spark mass (<1 A.U.). Where the spark mass exceeded ~10 A.U., higher counts of IP3R1 puncta were observed consistently. Considered together, these correlation trends support the role of RyR3 as the primary generator of the Ca^2+^ spark and IP3R1 either as primers or amplifiers of the RyR3-mediated spontaneous Ca^2+^ release, observed previously in cardiac muscle [32].

The 10× EExM and dSTORM images (Figure 1), however, revealed RyR3 and IP3R1 subplasmalemmal organisation patterns were heterogeneous and less well-defined clusters compared with the RyR3 clusters observed in cardiac muscle previously [31]. The sites where sparks were recorded (i.e., the combined perimeter of regions demarcated by the FWHM of each spark) constituted ~28% of the full basal 2D surface area of the cells (*n* = 13 cells from five animals) and, in total, reflected subplasmalemmal zones where spontaneous Ca^2+^ release activity occurred. The spatial density of both RyR3 and IP3R1 puncta in these zones where Ca^2+^ sparks were recorded was 3.8–5.0 times higher than in the overall cell-wide density measured in the correlative dSTORM data (Figure 3E). The nearest neighbour distance between puncta within the spark sites followed a narrower and left-shifted distribution (Figure 3F-main panel) for both RyR3 (red; mean ± SD: 126.2 ± 95.3 nm) and IP3R1 (purple; 112.4 ± 86.2 nm) compared with the cell-wide global distributions (inset; 189.5 ± 160.1 nm and 179.4 ± 79.3 nm, respectively). The distribution of RyR3 nearest neighbour distance within the spark site was marginally wider compared with that of IP3R, supporting the idea that the subplasmalemmal regions with visible RyR3 puncta clusters (e.g., annotated in Figure 1J,K) are likely origins of the Ca^2+^ sparks.

## 4. Discussion

### 4.1. A Correlative Approach to Probing the Nanoscale Organisation of RyR3 and IP3R1 Underlying Ca^2+^ Sparks

We have adapted a correlative imaging protocol recently developed for imaging Ca^2+^ sparks from primary cell types against super-resolution maps of the Ca^2+^ channels of origin [24]. Pragmatically, this approach has allowed us to access a nanoscale structure–function analysis in cell types such as DRG without the requirement for animal models with genetically encoded Ca^2+^ channel tags and/or Ca^2+^ sensors. We were able to carry out these experiments with existing Ca^2+^ indicators (Fluo-4 AM) and well-established dSTORM protocols. It has shown a spatially resolved functional correlate (i.e., Ca^2+^ sparks) to interrogate the nanoscale patterns of subplasmalemmal RyR3 and IP3R1 organisation. In addition to the use of channel-specific pharmacological agents to isolate Ca^2+^ release phenotypes, the experiments and analysis presented here can become a useful tool for probing the location-specific activity of individual or nanoscale organisation of signalling proteins.

### 4.2. Subplasmalemmal Expression Patterns of RyR3 and IP3R1 and Role in the Genesis of Ca^2+^ Sparks

The 10× EExM and the dSTORM image data together show punctate labelling densities of RyR3 in the subplasmalemmal spaces of the DRG somata. These puncta are distinctly smaller than the clustered morphologies produced in confocal images shown by Ouyang et al. [4]. In fact, the puncta area analyses for both 10× EExM and dSTORM data show that >95% of the RyR3 puncta occupy an area no larger than 2000 nm^2^ (Appendix A) which, on the approximation of a 30 × 30 nm square lattice array, is equivalent to ≤3 RyR3s. Simulation of the confocal images from super-resolution data has shown that whilst some clusters are recognisable, many others are below the sensitivity of confocal images (Appendix A). Given the size range of 30–50 nm of individual RyR3 puncta resolved in these super-resolution images and the true size of the RyR3 tetramer measured at ~34 Å [33], we postulate that these fine labelling densities reflect either individual channels or small groupings of yet-unresolved channels. The loose clustering of the puncta (over sub-micron length scales) likely reflects more distinct signalling nanodomains that bring RyR3s or small RyR3 clusters closer to plasmalemmal triggers such as L-type Ca^2+^ channels facilitated by structural proteins like junctophilin-3 and -4 [34]. The larger sizes of these loose clusters may reflect the ER cisterns that form membrane contact sites (MCSs), visualised with focused ion beam-scanning electron microscopy [35].

IP3R1 super-resolution images also report a punctate morphology in both 10× EExM and dSTORM, closely reproducing earlier observations with STORM [18]. Whilst there were no visually discernible clusters in these images over longer length scales (e.g., >200 nm), the correlative data that include maps of Ca^2+^ spark sites revealed approximately a five-fold higher density and ~40% shorter nearest neighbour distances in the IP3R1 organisation. A similarly higher density of RyR3 in the spark sites is consistent with a notable co-clustering of the two types of ER Ca^2+^ release channels in zones with a high degree of spontaneous Ca^2+^ spark activity (Figure 2D,E). The overlays of the spark centroids with the correlative dSTORM image showed broad correlation with areas with high densities of IP3R1 and RyR3. However, many spark centroids did not directly align with discernible clusters or puncta. It is likely that the origins of Ca^2+^ sparks lie between either structural or functional co-clusters of RyR3 and/or IP3R1 (similar to the ‘functional super-clusters’ seen in other cell types [29]).

The scattergrams of spark mass against the numbers of RyR3 and IP3R1 demonstrated a stronger correlation with the underlying RyR3 puncta count, supporting the role of RyR3 as the primary source of Ca^2+^ in these sparks. The analysis also showed that sparks often coincide with relatively modest numbers of RyR3 puncta, but a greater range of IP3R1 puncta. If the individual RyR3 puncta constitute a relatively fixed number of RyR3 channel clustering as argued above (e.g., ≤3 RyR3s, as suggested by the puncta size analysis in Appendix A), this correlation trend supports the idea that the amount of Ca^2+^ released in the spark is principally modulated by IP3R1. The shift in the spark mass with no change to the spark width in the presence of the indirect IP3R1 stimulator, bradykinin, suggests that this modulation is provided to a relatively fixed local population of RyR3s. This reinforces our above idea of functional super-clusters that are encoded in the higher density of RyR3 and IP3R1 organisation within the spark sites.

IP3Rs may impart this modulatory effect on the genesis of sparks by increasing the sensitivity of RyR3 to opening by elevating the local cytoplasmic Ca^2+^. In cardiac muscle, Ca^2+^ actually increases the frequency of the sparks, but leads to a reduction in amplitude [36]; however, if the free cytoplasmic Ca^2+^ concentration [Ca^2+^]_i_ is well buffered, the opposite effect is to be expected. In DRG, local [Ca^2+^]_i_ is likely to also be influenced by other Ca^2+^ sources such as L-type Ca^2+^ channels and the Ca^2+^-release-activated Ca^2+^ channel complex (STIM1-Orai1); the components of this complex are clustered into signalling nanodomains via structural proteins such as junctophilins known to exist in the DRG ER membrane [20,34]. Together, it is possible that these Ca^2+^ sources make up subplasmalemmal signalling hubs that amplify each other’s activation in order to sensitise the cell towards noxious stimuli.

The functional role of RyR3-mediated calcium release can also vary depending on the location of the neuronal soma. For example, axonal growth cones of neuronal somata in the dorsal grey commissure were impaired by RyR3 knockout whilst those in neurons in the superficial dorsal horn were unaffected [37]. Further, numerous transcriptomic studies that have been published since the conclusion of our study illustrate classification of subpopulations of DRG neurons with divergent expression patterns of key genes that could represent distinct nociceptive specialisation [38,39,40]. Future investigations into the functional impact of RyR3-mediated calcium sparks or the expression patterns of key calcium handling proteins (with techniques like ExM or dSTORM) must therefore account for possible variations between DRG subpopulations.

### 4.3. Limitations and Uncertainties

A number of limitations are noteworthy in our experimental approach. Deviations in the assumed flatness of the cell surface, asymmetries and heterogeneities in the 2D spread of Ca^2+^ in the recorded sparks and any acute remodelling or mobility of the channels cannot be ruled out entirely. Whilst the Ca^2+^ sparks detected in TIRF images were filtered by width and the localisation confidence value (*R*^2^), we were unlikely to be able to eliminate sparks originating in the deeper interior of the cytoplasm rather than in the TIRF illuminated subplasmalemmal regions. Our approach of fast, 2D imaging of Ca^2+^ sparks across the subplasmalemmal regions, on face value, could also over-represent the spatial density and frequency of sparks compared with data recorded by Oyuang et al. [4]. This is because the finer axial integration (<200 nm in TIRF, compared with >600 nm in the confocal point scanning used by them) and lack of light rejection via a pinhole allow greater contrast in the calcium images and sensitivity in detecting events of lower fluorescence intensity. The higher density of Ca^2+^ sparks at the cell periphery compared with the centre of the cells [4] can also allow the observed spark density to exceed the true density of Ca^2+^ sparks throughout the cell without any changes to the physiological condition of the cells.

In the cocultures, DRG somata were often observed to be in contact with the surrounding glial cells, which limited the population of DRG neurons accessible for illumination with the TIRF evanescent field. It was therefore critical to analyse only the DRG neurons’ subplasmalemmal regions located within the shallow TIRF field in both the calcium spark imaging and dSTORM analysis. Whilst we relied upon a neuronal and glial coculture protocol optimised for neuronal calcium imaging previously, the use of a satellite glial cell inhibitor such as Ara-c could perhaps have helped achieve more straightforward selection of DRG neurons and characterisation of RyR3/IP3R1 expression patterns with ExM.

The potentiation of the spark mass when exposed to caffeine and bradykinin separately underscored the roles of RyR and IP3R1 in spark genesis. In the correlative Ca^2+^ spark experiments, cells immersed in Tyrode’s solution with 5 mM [Ca^2+^]_o_, however, lacked these agonists. Under these conditions, the contribution of trans-plasmalemmal Ca^2+^ fluxes into the subplasmalemmal space cannot be ruled out. Repeated or sequential calcium imaging protocols are prohibitive in sequential correlative imaging experiments in approaches similar to ours. However, imaging Ca^2+^ sparks under 0 [Ca^2+^]_o_, ideally in the presence of either caffeine or bradykinin, would have verified the source of these events as truly intracellular.

Furthermore, the efficiency of RyR3 and IP3R1 labelling with the use of widely established antibodies is likely to be less than 100%, in spite of using antibodies that have been previously characterised for fluorescence microscopy. The incompatibility of the antibodies (due to the same species of origin) with two-colour dSTORM prevented us from making direct observations on the mutual co-clustering patterns of RyR3 and IP3R1.

Whilst the correlative analysis would have helped decode more directly the Ca^2+^ spark phenotype unlocked by local interaction of the channels, the identities of the RyR3 and IP3R1 puncta still need to be established in terms of the true number of underlying channels. Inherent to the protocol of sequential live-cell calcium and dSTORM imaging is the likelihood of IP3R1 or RyR3 mobility within the time window of Ca^2+^ spark imaging and fixation prior to dSTORM. In spite of reports of IP3R1 and RyR mobility in the ER of HEK293 [41] cells and cardiomyocytes [42], respectively, we are not aware of reports characterising this feature in DRG. The precision of the correlative imaging approach is ultimately limited by the alignment error between the averaged (diffraction-limited) Ca^2+^ image and the super-resolution image. The key determinants of the latter are the resolution and the signal-to-noise ratio (SNR) of the averaged Ca^2+^ image. Appendix A outlines a series of simulations by which we demonstrate an alignment error under 100 nm for each iteration given the typical TIRF resolution of ~250 nm and SNR of ~17.0 in our averaged Ca^2+^ image data. Finally, the correlative approach also prevented us from calibrating the pixel values in the Ca^2+^ images against [Ca^2+^]_i_ with a Ca^2+^ ionophore due to the requirement of preserving the cellular structure for dSTORM. As a standardisation, all experiments were performed under identical excitation intensities and camera settings, which allowed us to quantitatively compare the intensity information in Ca^2+^ images between cells and samples.

## Figures and Tables

**Figure 1 cells-13-00038-f001:**
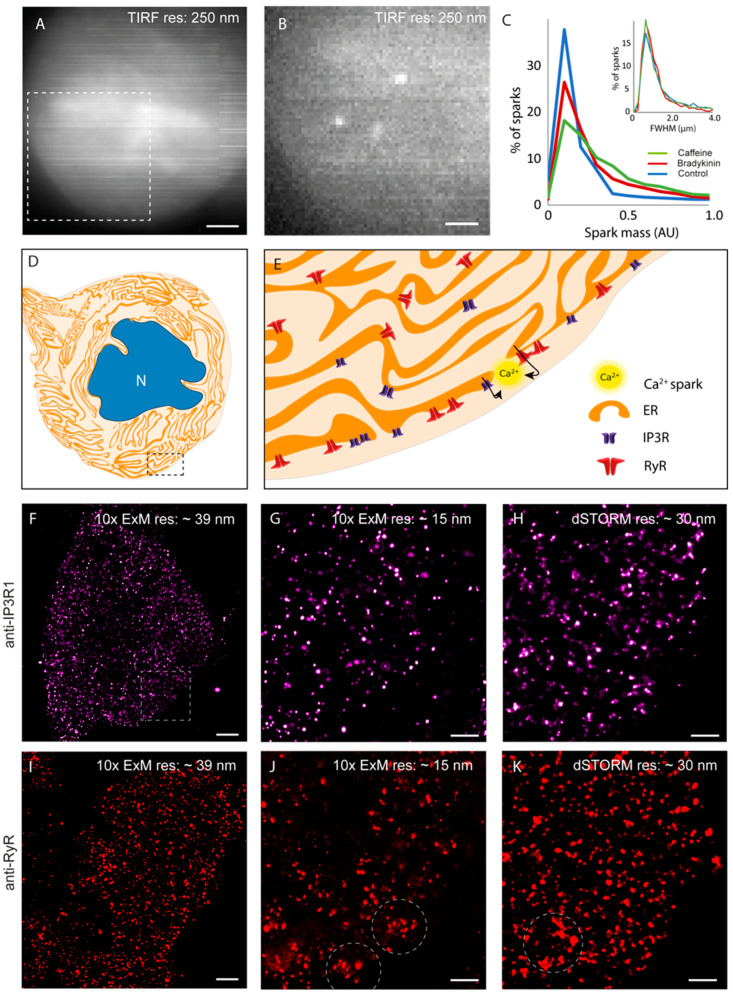
Observing Ca^2+^ sparks, RyR3 and IP3R in dorsal root ganglion neurons. (**A**) A TIRF image of a cultured DRG neuronal soma loaded with Fluo-4AM (resolution ~250 nm). (**B**) Magnified view of the dashed box region in panel A where spontaneous Ca^2+^ sparks were recorded in the absence of either caffeine or bradykinin under camera integration time of 50 ms. (**C**) Overlaid percentage histograms show an increase in the integrated spark mass (shown in arbitrary units) of spontaneous Ca^2+^ sparks in both cells treated with bradykinin (red) and cells exposed to caffeine (green) compared with controls (in the absence of caffeine or bradykinin), respectively. The equivalent percentage histogram for FWHM of the sparks (inset) shows that it remains unaltered in the presence of bradykinin and caffeine. Spark data pooled from 5 cells in each condition, cells sourced from 5 animals, with the same cell being exposed to each treatment as a repeated measure. Histogram bin sizes: 0.1 for spark mass, 0.2 μm for FWHM. (**D**) The cartoon illustrates the extensive ER in the DRG soma which occupy the cytoplasm. *N* notates the nucleus. (**E**) It is hypothesised that the Ca^2+^ sparks which are observed frequently at the subplasmalemmal regions (shown in zoomed view of the cartoon in the dashed box in (**D**)) may arise from either RyR3 (red) and/or IP3R (purple). (**F**) IP3R1 immunofluorescence near the bottom surface of the cell visualised with 10× ExM combined with confocal microscopy (with a 20× 0.9 NA objective), at an applied resolution of ~39 nm. (**G**) The region indicated by the region in dashed box in (**F**) is shown, imaged with a 40× 1.3 NA lens (resolution ~15 nm) illustrating a highly nonuniform distribution of punctate labelling densities. (**H**) This morphology was qualitatively reproduced with dSTORM images of equivalent regions (resolution ~30 nm). (**I**) The RyR3 immunofluorescence imaged with 10× ExM, combined with confocal microscopy (with a 0.8 NA lens) at a resolution of ~39 nm. (**J**) A similar cell from the same sample imaged in Airyscan mode with a 1.3 NA objective (effective resolution ~15 nm) shows a heterogeneous and punctate labelling morphology. Observed were loose clustering patterns at distinct sub-micron-scale domains of the membrane (dashed circles). (**K**) An equivalent dSTORM image reproduced the same morphologies, including the loose clusters. Scale bars: (**A**) 5 μm, (**B**) 3 μm, (**F**,**I**) 1 μm, (**G**,**H**,**J**,**K**) 500 nm.

**Figure 2 cells-13-00038-f002:**
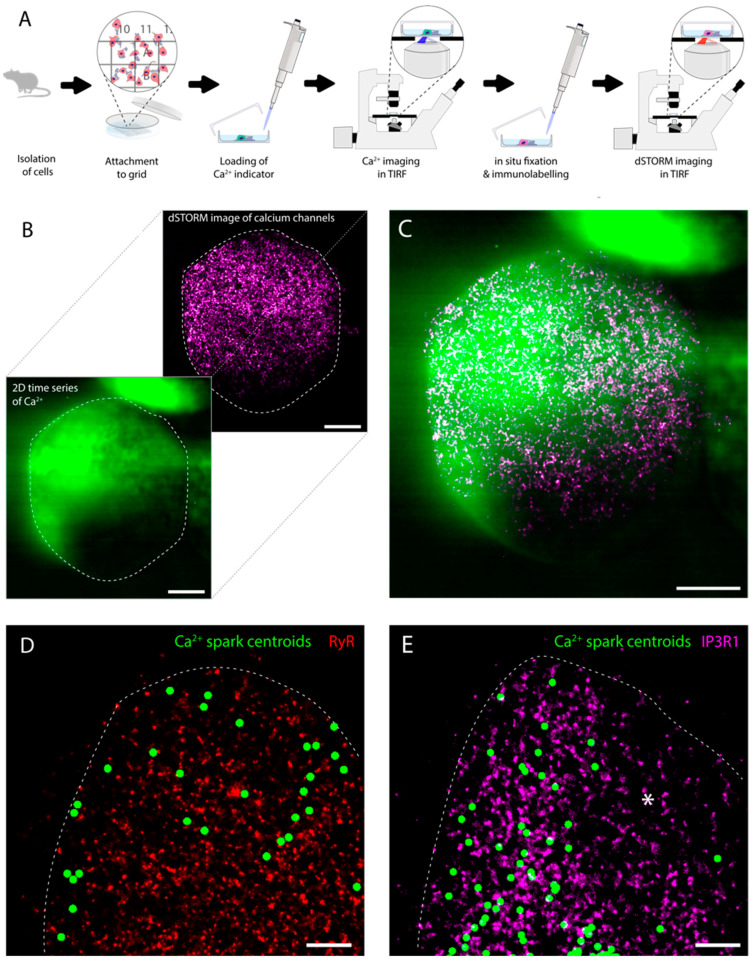
The experimental pipeline of correlative TIRF Ca^2+^ spark and dSTORM imaging. (**A**) The key steps include extracting the DRG neurons from the neonatal rats, dissecting, isolating and coculturing of DRG and glial cells within the gridded dishes, AM-loading of the Ca^2+^ indicator dye and de-esterification, recording Ca^2+^-sensitive indicator dye fluorescence from DRG neuronal cells and the specific grid coordinates noted, fixation and immunofluorescence labelling and returning to the coordinates of a cell whose Ca^2+^ signals were recorded previously to acquire dSTORM data of RyR3 or IP3R. (**B**) The post-acquisition alignment of the averaged and scale-matched Ca^2+^ frame TIRF image (green) to the rendered dSTORM image (IP3R image; purple) through a semiautomated programme chiefly relying on the cell outline (indicated with white dashed lines). (**C**) Shown is an overlay of the two images in their respective colour tables. (**D**,**E**) Overlays of the centroids of spontaneous Ca^2+^ sparks (green) recorded in the absence of either caffeine or bradykinin during a 10 s time window and dSTORM images of RyR3 (red) and IP3R1 (magenta), respectively. Asterisk in panel E denotes regions with a glancing view of the cell surface with lower density of calcium sparks observed. Scale bars: (**B**,**C**) 5 μm, (**D**,**E**) 1 μm.

**Figure 3 cells-13-00038-f003:**
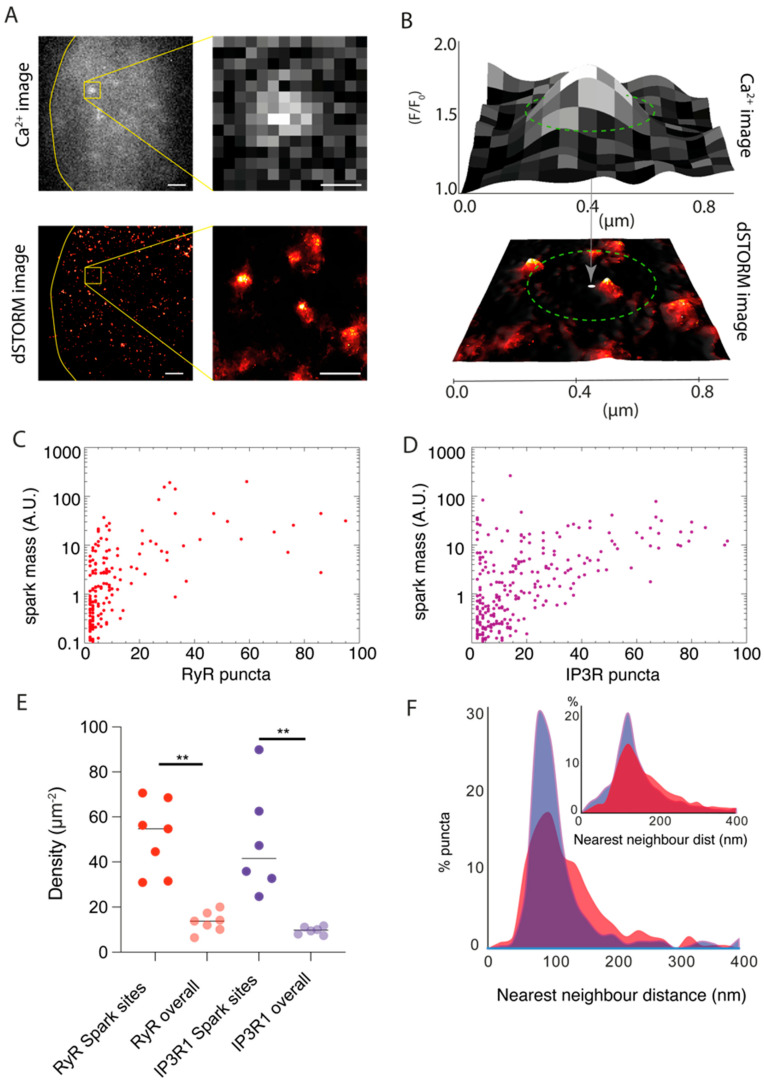
Local correlation of spontaneous Ca^2+^ sparks and localisation patterns of RyR3 and IP3R1. (**A**) Shown are the TIRF frame (grey colour table) of spontaneous Ca^2+^ sparks in a DRG neuron in the absence of caffeine or bradykinin and the local RyR3 labelling pattern in the same cell (red). The yellow box shows the magnified view of the sparks and the punctate RyR3 pattern locally. (**B**) In the local correlation analysis, the FWHM (green dashed line in upper panel) of the localised Ca^2+^ spark provided a window of confidence (green dashed lines in lower panel) to examine the dSTORM RyR3 puncta in the region of origin for the Ca^2+^ spark. (**C**) Scatterplot between the integral of the Ca^2+^ released (estimated based on ‘spark mass’, shown in arbitrary units) and the number of RyR3 puncta within the interrogation window shows a majority of sparks coinciding with small groups of RyR3 puncta (<10 puncta). (**D**) Scatterplot between the spark mass and the local IP3R puncta count shows sparks with larger integrals can coincide with a broader range of total IP3R puncta. (**E**) Dot plot (mean marked with solid line) showing that the densities of both RyR3 and IP3R puncta (per μm^2^ of 2D image area) were higher in spark sites (51.1 μm^−2^ and 48.8 μm^−2^) compared with overall footprint of the cell (13.4 μm^−2^ and 9.64 μm^−2^). ** Tukey test results for RyR3 and IP3R comparisons indicated the following: q_s_ = 6.89 and 6.65, respectively; *p* = 0.0004 and 0.0006, respectively; df = 22 and 22. (**F**) Overlaid percentage histograms of the nearest neighbour distances of RyR3 (red) or IP3R (purple) puncta within the spark sites. The inset shows the equivalent histograms of RyR3 and IP3R puncta in the cell area overall. Scale bars: (**A**) left panels: 1 μm, (**A**) right panels: 250 nm.

## Data Availability

Datasets and code from the RyR3 and IP3R correlative analysis have been deposited on an online repository which will be linked with the https://zenodo.org/doi/10.5281/zenodo.10157969, accessed on 20 December 2023. The general source code for xyspark and the correlative image alignment can be accessed via the xyspark (https://xyspark.leeds.ac.uk/, accessed on 20 December 2023) and GitHub (https://github.com/ijayas/imagealigning/, accessed on 20 December 2023) repositories, respectively.

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
