# Peer review of "Super-Resolution Analysis of the Origins of the Elementary Events of ER Calcium Release in Dorsal Root Ganglion Neurons"

_cells, 2023, doi:10.3390/cells13010038_

Round 1

Reviewer 1 Report (Previous Reviewer 2)

Comments and Suggestions for Authors

The authors have addressed the comments adequately, except the following:

"4. The rate of spontaneous activity seems to be quite high, but this topic is not discussed. What can trigger this activity?

We thank the reviewer for this observation and caution on comparing methodological differences that can lead to differences in the calcium spark density. Comparisons we draw with the study of Oyuang et al paper (PMCID: PMC1189299) must take into account two key differences in the way calcium sparks were recorded. Ouyang et al record calcium sparks with confocal (line scan and two-dimensional point scan) microscopy which inherently features an axial signal integration of > 600 nm. Our approach used TIRF microscopy with <200 nm axial integration and no light rejection (due to the absence of a pinhole) which are likely to allow greater contrast and sensitivity in detecting sparks. Secondly, line scans used by Oyuang et al were placed across the centre of the cell whilst the TIRF fields that we imaged recorded exclusively within the sub-plasmalemmal regions where the frequency of sparks are inherently higher (see Figure 1-C of Oyuang et al).

We have added a paragraph to the Discussion section 4.3 of the revised manuscript outlining the differences between studies (page 12, lines 548-556)"

 The point was not whether the results are an overestimate or not. DRG neurons are a biological system, and the signals described here are a biological phenomenon. My question is: what evokes the calcium sparks?

Author Response

The authors have addressed the comments adequately, except the following:

 "4. The rate of spontaneous activity seems to be quite high, but this topic is not discussed. What can trigger this activity?

We thank the reviewer for this observation and caution on comparing methodological differences that can lead to differences in the calcium spark density. Comparisons we draw with the study of Oyuang et al paper (PMCID: PMC1189299) must take into account two key differences in the way calcium sparks were recorded. Ouyang et al record calcium sparks with confocal (line scan and two-dimensional point scan) microscopy which inherently features an axial signal integration of > 600 nm. Our approach used TIRF microscopy with <200 nm axial integration and no light rejection (due to the absence of a pinhole) which are likely to allow greater contrast and sensitivity in detecting sparks. Secondly, line scans used by Oyuang et al were placed across the centre of the cell whilst the TIRF fields that we imaged recorded exclusively within the sub-plasmalemmal regions where the frequency of sparks are inherently higher (see Figure 1-C of Oyuang et al).

We have added a paragraph to the Discussion section 4.3 of the revised manuscript outlining the differences between studies (page 12, lines 548-556)"

 The point was not whether the results are an overestimate or not. DRG neurons are a biological system, and the signals described here are a biological phenomenon. My question is: what evokes the calcium sparks?

We thank the reviewer for the positive assessment of our revised manuscript. We are also grateful for clarifying this question. In our first response, we misunderstood the reviewer’s assessment of the rate of spontaneous calcium spark activity being quite high as an issue that reflected upon the different sensitivities of the spark imaging modalities.

In answer to the clarified question, our experimental approach confirms the report of Oyuang et al (PMCID: PMC1189299) that DRG neuronal calcium sparks can either be evoked by the administration of stimulants of RyR or IP3R1 (i.e. caffeine or bradykinin) or occur spontaneously. Spontaneous calcium release events in isolated neurons and axonal terminals and spines (such as ‘calcium syntillas’ and calcium puffs) in the presence of local depolarisations have previously been reported (e.g. PMCID: PMC2810286) . Oyuang et al conclude that spontaneous sparks in DRG soma would rely upon calcium influxes via voltage-gated calcium channels such as L-, N-, Q-, and T-type channels. This is supported by our experimental conditions of millimolar-level [Ca2+]o required for calcium spark imaging. However, we cannot rule out the likelihood that the DRG neurons, as a system accommodating to high [Ca2+]o, can reach high intra-ER luminal [Ca2+] and ensuing store-driven calcium leak that would either (i) prime individual RyR channels for plasmalemmal triggers, or (ii) recruit local RyR, via calcium-induced-calcium-release, to produce coordinated calcium sparks. To our knowledge, these possibilities remain to be systematically studied.

Reviewer 2 Report (New Reviewer)

Comments and Suggestions for Authors

The manuscript by Hurley et al. examines the role of RyR3 and IP3R1 in generating calcium sparks from the endoplasmic reticulum (ER) contact sites with plasma membrane (PM), in DRG neurons. Here a recently developed technology and pipeline previously published by the authors was used to detect calcium sparks and correlate the spatial distribution of calcium sparks with RyR3 and IP3 expression.

While the experiments appear to have a sound technical basis, concerns may be raised about the novelty of information raised here about DRG pathophysiology. The work is interesting but, according to this reviewer, some information is needed to reach the authors' conclusions.

Major concerns:

1.      The culturing conditions are not clear; why the authors did not use Ara-c if satellite glial cells were interfering with the imaging? Also is not clear why they use non-physiological calcium concentration. For instance, the DRG medium was replaced with Tyrode’s solution after 48 hours which contains 0,75 mM (physiological values are 1,8-2,0 mM). For how much time did the cells remain in this condition without a DMEM-based medium? Also, the authors should clarify why they use 5 mM Ca++ to detect the sparks. Were they able to detect sparks in a 2 mM extracellular calcium condition? Moreover, I would have tried to treat DRG neurons with caffeine or bradykinin after a quick washout of extracellular calcium to be sure they were detecting intracellular events. Did the authors perform these experiments? If not they should at least discuss it and improve the method section.

2.      I am not sure if 5 mM caffeine would be the best way to study single calcium sparks. It is well known in the field that such concentration triggers global calcium events. Did the authors try sub-millimolar caffeine concentrations as previously reported?

3.      Why spark mass does not correlate with the number of RyR3 puncta in the spark region? Even though it is clear that the sparks broadly correlate with the receptor density,  I would expect a better correlation between the mass and the number of RyR3, given that caffeine 5 mM should virtually activate all the receptors and assuming that sparks constitute the building blocks of the global Ca++ transients and based on unitary properties of sparks. Can the authors explain this discrepancy?

4.      In the introduction the authors stated that  RYR and IP3R are critical in DRG calcium signaling and pain perception. The authors should tone down these statements. There are still conflicting results about the expression of RyR in DRG: there are few reports which observed that receptors 1 and 3 are expressed (PMID: 20089138), only RyR3 in DRG microsomes (PMID: 11821055) and all 3 isoforms (PMID: 28899787). Moreover, there are no convincing studies that correlate RyR receptor loss or gain of function with pain. RyR has been associated with the axonal growth cone but knocking out RyR affected only on the axons reaching deep layers of the dorsal horn and not superficial layers (laminae I and II) which account for pain transmission and derive from TRPV1 positive cells (PMID: 16172206);

5.      There are convincing papers that observed that only a small proportion of DRG  can be triggered by caffeine or IP3 agonists. The Gold’s lab performed a very systematic study on calcium transients evoked by depolarization and observed that CICR via RyR contributes to calcium transients in large and medium DRG insensitive to Capsaicin (PMID: 16172206); I am not sure how important is the contribution to RyR in pain states as stated in many part of the current manuscript. This is also similar to IP3R (PMID: 9189891, 16172206). The authors should discuss this point with the existing literature, especially if they found calcium sparks in every DRG.

6.      Linked to the previous point: DRGs have different sub-populations and thanks to transcriptomic studies we know that the picture is more complicated than thought before. Did the authors perform these experiments for instance on TRPV1 positive DRG? If not this should be indicated in the discussion (is a limitation) and methods.

Minor:

1.      I would refer to RYR3 throughout the text since the authors use a selective RyR3 antibody;

2.      Figure 2: are the Ca++ sparks spontaneous or induced? Please clarify;

3.      See above: please clarify throughout the figures if sparks were spontaneous or obtained under stimulation (caffeine or bradykinin);

4.      202: did the authors also perform the same experiments on satellite glial cells? Also, it should be important to discuss this point in the discussion.

Author Response

Reviewer 2:

  1. The culturing conditions are not clear; why the authors did not use Ara-c if satellite glial cells were interfering with the imaging? Also is not clear why they use non-physiological calcium concentration. For instance, the DRG medium was replaced with Tyrode’s solution after 48 hours which contains 0,75 mM (physiological values are 1,8-2,0 mM). For how much time did the cells remain in this condition without a DMEM-based medium? Also, the authors should clarify why they use 5 mM Ca++to detect the sparks. Were they able to detect sparks in a 2 mM extracellular calcium condition? Moreover, I would have tried to treat DRG neurons with caffeine or bradykinin after a quick washout of extracellular calcium to be sure they were detecting intracellular events. Did the authors perform these experiments? If not they should at least discuss it and improve the method section.

We have updated the text to include more details of the culture protocol (Methods section, page 3, lines 105-119) which has been optimised in previous studies (e.g. PMCID: PMC6087463) for DRG-specific calcium imaging experiments.

As outlined in our previous set of responses to reviewer comments, we did not find the glial cells to be interfering with the imaging. DRGs were distinguishable from glial cells on the basis of their distinct morphology and size. The calcium sparks were recorded with a TIRF field which ensured that their origins were very much within the DRG cells in focus, and not interfaces between DRGs and glial cells. Because we chose to rely on the selectivity of the field of detection of calcium sparks with TIRF allowed us to sample DRG calcium sparks selectively, we did not see the need to inhibit glial cells from the cultures. We have added a statement on this limitation to the Discussion section (page 12, line 598-603)

A variety of calcium concentrations in Tyrode’s solution were applied to the cells in the short window of calcium imaging, with 5 mM selected on the basis that cells were able to withstand this extracellular concentration. We acknowledge that sequential washout of calcium and exposure to caffeine and bradykinin would have allowed us to exclude any contributions from plasmalemmal calcium fluxed. We have added this point to the Discussion (see page 12, line 604-611).

  1. I am not sure if 5 mM caffeine would be the best way to study single calcium sparks. It is well known in the field that such concentration triggers global calcium events. Did the authors try sub-millimolar caffeine concentrations as previously reported?

We have noted the range of different concentrations of caffeine used by other groups previously. For this reason, we tried caffeine concentrations ranging between 0.15 and 10 mM. To show the impact of caffeine on the calcium spark size, we needed to establish a supra-maximal concentration of caffeine that would show any shifts in spark width and/or spark mass. As shown in Figure 1C, we observed no change in spark width, however 5 mM was the lowest concentration at which caffeine showed the ~ 60% increase in the spark mass. At 10 mM, this effect was largely unchanged from 5 mM.

  1. Why spark mass does not correlate with the number of RyR3 puncta in the spark region? Even though it is clear that the sparks broadly correlate with the receptor density,  I would expect a better correlation between the mass and the number of RyR3, given that caffeine 5 mM should virtually activate all the receptors and assuming that sparks constitute the building blocks of the global Ca++transients and based on unitary properties of sparks. Can the authors explain this discrepancy?

The scatterplot (Figure 3C) shows calcium sparks and RyR distribution in cells not exposed to caffeine. The objective of the study was to study the relative contributions of RyRs and IP3R1s to the genesis of subplasmalemmal calcium sparks. Caffeine treatment would potentiate RyRs and certainly alters the recruitment of RyRs to the genesis of calcium spark; it clearly shifts the spark mass upwards (as shown in Figure 1C) and would therefore create an artificial correlation.

The weak correlation between RyR3 puncta count and the spark mass in Figure 3C comes from the sparsity of RyR channel organisation (totalling ≤3 RyR) in the vast majority of the regions where calcium sparks were recorded. With such small densities of channels, and the small spark masses (Figure 1C; typically 1-2 orders of magnitude smaller than sparks in cardiac muscle), it is unlikely that we will see this correlation over such a narrow range of RyR groupings. However, we do see a greater range of IP3R1 puncta counts across the spark sites and a stronger correlation with spark mass which is consistent with the idea of IP3R1 as the principal modulator of spark mass.

  1. In the introduction the authors stated that  RYR and IP3R are critical in DRG calcium signaling and pain perception. The authors should tone down these statements. There are still conflicting results about the expression of RyR in DRG: there are few reports which observed that receptors 1 and 3 are expressed (PMID: 20089138), only RyR3 in DRG microsomes (PMID: 11821055) and all 3 isoforms (PMID: 28899787). Moreover, there are no convincing studies that correlate RyR receptor loss or gain of function with pain. RyR has been associated with the axonal growth cone but knocking out RyR affected only on the axons reaching deep layers of the dorsal horn and not superficial layers (laminae I and II) which account for pain transmission and derive from TRPV1 positive cells (PMID: 16172206);

We acknowledge the reviewer’s concern about the conflicting reports on the expression of RyR types 1 to 3 in DRGs. We have rephrased the relevant section of the introduction to tone down the putative role of RyRs in modulating nociceptive signals (page 2, lines 62-66). We have also added a sentence to the Discussion to raise the uncertainty of the functional role of RyR3 in DRG function in a location-specific manner (see page 12, lines 461-466).

  1. There are convincing papers that observed that only a small proportion of DRG  can be triggered by caffeine or IP3 agonists. The Gold’s lab performed a very systematic study on calcium transients evoked by depolarization and observed that CICR via RyR contributes to calcium transients in large and medium DRG insensitive to Capsaicin (PMID: 16172206); I am not sure how important is the contribution to RyR in pain states as stated in many part of the current manuscript. This is also similar to IP3R (PMID: 9189891, 16172206). The authors should discuss this point with the existing literature, especially if they found calcium sparks in every DRG.

The reviewer disputes a role of RyR in pain sensing in spite of numerous studies observing the presence of calcium sparks and calcium transients in DRG neurons and demonstrations of a link between calcium sparks and vesicular secretion (PMCID: PMC1189299). We acknowledge the reviewer’s point that the presence of a signalling mechanism or phenomenon does not directly link to the neurons’ nociceptive function, at least in the context of the evidence available.

There were only a handful of places in this paper where a link of RyRs/ calcium sparks had been drawn to nociception. We have rephrased these, including the Abstract (page 2, line 14), Introduction (page 2, line 66-71), and Discussion (page 12, line 567), to reinforce the putative nature of this functional link.

  1. Linked to the previous point: DRGs have different sub-populations and thanks to transcriptomic studies we know that the picture is more complicated than thought before. Did the authors perform these experiments for instance on TRPV1 positive DRG? If not this should be indicated in the discussion (is a limitation) and methods.

We recognise the reviewer’s point that the transcriptomic approaches, in addition to morphometric classifications, can bring valuable and likely crucial distinctions between functionally different DRG neurons particularly in pain processing phenotypes. Many of these studies have emerged since the conclusion of the study presented in this manuscript. We have highlighted this limitation in our study in the Discussion section (see page 12, lines 573-576).

Minor:

  1. I would refer to RYR3 throughout the text since the authors use a selective RyR3 antibody

Done

  1. Figure 2: are the Ca++sparks spontaneous or induced? Please clarify;

The calcium sparks mapped here are all spontaneous events. We have revised the figure legend to make this clearer.

  1. See above: please clarify throughout the figures if sparks were spontaneous or obtained under stimulation (caffeine or bradykinin)

Done.

  1. 202: did the authors also perform the same experiments on satellite glial cells? Also, it should be important to discuss this point in the discussion.

We performed the expansion microscopy on DRG/glial cell co-cultures. DRG neurons were selectively imaged on the basis of their large size. We have incorporated this point into the new Discussion point on this limitation (page 12, line 595-600).

Round 2

Reviewer 2 Report (New Reviewer)

Comments and Suggestions for Authors

The manuscript has been well improved, and I think most of my comments have been addressed with necessary discussions.

This manuscript is a resubmission of an earlier submission. The following is a list of the peer review reports and author responses from that submission.

Round 1

Reviewer 1 Report

Comments and Suggestions for Authors

The manuscript entitled: ‘’Super-Resolution Analysis of the Origins of the Elementary Events of ER Calcium Release in Dorsal Root Ganglion Neurons’’ by Hurley et al. presents usage of novel correlative microscopy to visualize the distribution of ryanodine and IP3 receptors in dorsal root ganglion neurons in vitro and a correlation of the receptors’ distribution with fluorescently-detected local calcium sparks. I found the manuscript very interesting and promising, however, the study cannot be published in its current form due to concerns I listed below. I divided my concerns into major and minor:

Major concerns:

1.       The authors in the Material and Methods section did not describe the condition of dorsal root ganglion neurons culturing. One cannot say for how many days the neurons were cultured before the experiment and if the culture medium was supplemented (doi: 10.3791/57569). It is stated in the text that the “co-cultures” were used – it suggests that no antimetabolite was added. Knowing that the co-cultures of unknown DIV were used, it cannot be excluded that the observed properties of the receptors are caused by the cross-talk between neuronal and non-neuronal cells (Schwann and satellite cells) but these are not an immanent feature of the dorsal root ganglion cells. Also no prove has been presented that the cells took into analysis are indeed the dorsal root ganglion neurons of a particular type – the properties of endoplasmic reticulum of in situ unmyelinated C-fiber neurons may differ from the other types of dorsal root ganglion neurons. Additionally, authors themselves pointed out the limitations of the in vitro model used;

2.       On the other hand, it can be assumed that the aim of the study was to present a novel technique of microscopic visualisation instead of characterising the correlation between calcium receptors distribution and calcium sparks in dorsal root ganglion neurons. Regarding that, it has to be noticed that the manuscript loses its novelty because the protocol presented within is already published by the same authors (doi.org/10.1098/rsob.230045). The current manuscript contains the very same set of data as already published, with the only difference being the type of the cells used – here neurons, there cardiac muscle. I my opinion, a simple recapitulation of results obtained via a protocol published in an article in just another type of cells is not enough to be accepted for publication in the journal of such an impact;

3.       Calculating the percentage of integrated spark mass (Fig. 1C), the data were pooled from 5 cells in each condition (there are 3 conditions), and the cells were sourced from 5 animals. What does it mean? It means that to find 15 cells 5 animals were sacrificed? In such a case I would be afraid of condition and health of the analysed cells. Additionally, having data only from 5 cells per condition seems not to justify any obtained observation due to internal heterogeneity of cultures.

Minor concerns:

1.       In the main text some abbreviations are explained, whereas some are not. Each abbreviation used (protein or technique name) should be explained;

2.       Subchapters 2.3. and 2.5. are duplicated;

3.       The manuscript presents some numerical data and points out the differences. No statistics used is described in the Material and Methods section;

4.       The first paragraph of the subchapter 3.2. should be moved to Material and Methods section;

5.       Figures 3E and 3F are not mentioned in the text;

6.       Despite analysis neuronal cells, the authors sometimes describe sarcolemma instead of plasmalemma;

7.       Among 40 references listed in the bibliography, 9 of them are auto-citations. Whereas some of these seem to be justified due to the description of already published protocol and methods, some of them are not fully justified in my opinion.

Reviewer 2 Report

Comments and Suggestions for Authors

Calcium sparks are elementary events of calcium release from intracellular stores, and  are important in calcium dynamics. The topic of calcium sparks in the peripheral nervous system has received very little attention, and this paper is very timely. The authors used advanced microscopic methods to identify the calcium release sites. The methods are described in detail, and the supplementary material is helpful.  

Major comments

1. The cultures contain glial cells, which are mentioned several times, but their possible contributions to the recorded signals are not discussed.

"Areas similar to that asterisked in Fig 2E may arise from glial cell filopodia", (lines 281-2); what is the evidence for that?

It is not clear what is the "apical surface of glial cells" (line 436). A figure showing the arrangement of glial cells with respect to the neurons would be helpful.

2. The rate of spontaneous activity seems to be quite high, but this topic is not discussed. What can trigger this activity?

3. DRG neurons play crucial role in the transmission of sensory signals, including pain. However, the significance of the results to these functions is not discussed,

 Minor comments

Methods: Did you take fixation-induced shrinkage into account?

Figure legends need to be checked carefully and corrected, see below:

Fig. 1. "combine resolution" (?);  Line. 256 "observed"; what does "loose clusters" mean?

Fig. 2, "colour table" (?)

Fig. 3, "red hot" (?); "magnified region on the left", is this correct?;  "cell area" (?)

Writing

Overall, the writing is clear, but there are several expressions that need to be corrected. See for example:

Line 59 "higher localization" do you man density, concentration?

Line 38 "broadly observed"

Line 199.

Line 309, "Golgi and mitochondria".

Line 313 "Steep gradient? Between what?

Comments on the Quality of English Language

See above.